# Plant Serine Protease Inhibitors: Biotechnology Application in Agriculture and Molecular Farming

**DOI:** 10.3390/ijms20061345

**Published:** 2019-03-17

**Authors:** Marina Clemente, Mariana G. Corigliano, Sebastián A. Pariani, Edwin F. Sánchez-López, Valeria A. Sander, Víctor A. Ramos-Duarte

**Affiliations:** Instituto Tecnológico Chascomús (INTECH), UNSAM-CONICET, Chascomús, Provincia de Buenos Aires B7130, Argentina; marianacorigliano@intech.gov.ar (M.G.C.); sebapariani@hotmail.com (S.A.P.); esanchez@intech.gov.ar (E.F.S.-L.); valeriasander@intech.gov.ar (V.A.S.); varamosd@intech.gov.ar (V.A.R.-D.)

**Keywords:** serine protease inhibitors, plants, pathogen resistance, molecular farming

## Abstract

The serine protease inhibitors (SPIs) are widely distributed in living organisms like bacteria, fungi, plants, and humans. The main function of SPIs as protease enzymes is to regulate the proteolytic activity. In plants, most of the studies of SPIs have been focused on their physiological role. The initial studies carried out in plants showed that SPIs participate in the regulation of endogenous proteolytic processes, as the regulation of proteases in seeds. Besides, it was observed that SPIs also participate in the regulation of cell death during plant development and senescence. On the other hand, plant SPIs have an important role in plant defense against pests and phytopathogenic microorganisms. In the last 20 years, several transgenic plants over-expressing SPIs have been produced and tested in order to achieve the increase of the resistance against pathogenic insects. Finally, in molecular farming, SPIs have been employed to minimize the proteolysis of recombinant proteins expressed in plants. The present review discusses the potential biotechnological applications of plant SPIs in the agriculture field.

## 1. Introduction

Proteolytic enzymes (proteases, proteinases and/or peptidases) are involved in several proteolytic processes that regulate and avoid extreme and unnecessary protein degradation. This crucial process allows precise control of both the functionality and correct temporal-spatial location of proteins [1]. In animals, proteases play an important role in several cellular events such as inflammatory response, cellular apoptosis, blood coagulation, and hormone-processing pathways [2]. However, these enzymes can be also potentially harmful; therefore, the proteolytic activity should be accurately controlled.

The protease inhibitors (PIs) are essential for the regulation of proteolytic activity and play an important role in several biological processes related to metabolism and cell physiology [3]. In addition, in animals some PIs has been described as growth factors as well as receptors in cell-signaling pathways or during carcinogenesis processes [4]. Conversely, several inherited diseases such as emphysema or certain cases of epilepsy are due to the malfunction of specific PIs [5,6,7]. Therefore, PIs are indispensable components for cellular homeostasis and survival.

In plants, PIs also participate in many physiological processes [8,9]. They have been related to the mobilization of storage proteins, regulation of endogenous enzymatic activities, modulation of apoptosis and programmed cell death and stabilization of defense proteins or compounds against animals, insects and microorganisms [10,11,12,13,14,15,16,17]. Given their wide versatility and their broad spectrum of biotechnological applications, many plant PIs have been characterized [18]. This study presents a comprehensive revision of plant PIs and their biotechnological applications in the molecular agriculture field. This work also depicts the employment of PIs as a useful strategy to minimize the proteolysis of recombinant proteins in molecular farming. Having enormous potential in biotechnology applications, the scope for exploration of natural PIs remains wide open.

## 2. Classification of Protease Inhibitors

As is common in protein nomenclature, when a new PI is discovered it is named according to its biological origin and the enzyme that it inhibits (for example *Streptomyces* subtilisin inhibitor or pancreatic trypsin inhibitor) [19]. But this nomenclature does not allow for inferring either the relationship between the different inhibitors or whether the mechanism of inhibition for a particular inhibitor can be applied to others. Therefore, Laskowski and Kato [19] proposed classifying the PIs in families, considering the specific reactive site present in the sequences. This nomenclature made it possible to group PIs into four main families: (1) cysteine protease inhibitors, (2) metalloid protease inhibitors, (3) aspartic protease inhibitors, and (4) serine protease inhibitors. In plants, PIs are also classified according to their function of structural and biochemical properties, such as Bowman–Birk serine protease inhibitors, cereal trypsin/α-amylase inhibitors, cysteine protease inhibitors, metallo carboxypeptidase inhibitors, mustard trypsin inhibitors, potato type I inhibitors, potato-type II protease inhibitors, serpins, soybean trypsin (Kunitz) inhibitors and squash inhibitors [20,21].

Later, Rawlings et al. [3] proposed a new classification of PIs grouping them into families and clans. This classification is similar to the peptidases/proteases classification system proposed by Laskowski and Kato [19], but it aims to reflect the evolutionary relationships between PIs. This system has a hierarchical structure with three main levels: inhibitors, families and, clans [3]. The clan represents the highest level of evolutionary divergence. The sequences that belong to the same clan are evolutionarily related although they do not share high sequence similarity [22]. The proteins that belong to the same clan have similar tertiary structures. Simultaneously, clans are divided into families, which are grouped according to a common ancestor where all family members have similar aminoacidic sequences (homologous proteins) [23]. In summary, proteins that belong to the same family comprise related sequences, while proteins that belong to the same clan display related conformational structures. To determine to which family a PI belongs, an analysis along the protein sequence in the inhibitory region needs to be undertaken. This region is called “inhibitory unit” and it belongs to the PI domain that interacts with the protease domain. In some cases, the inhibitory unit may also contain the PI reactive site (P1). Therefore, the PI inhibitory unit corresponds to a structural domain, although there are protease inhibitors that contain more than one inhibitory domain [24].

In general, PIs from the same family inhibit a single catalytic type of protease using a similar mechanism. However, there are some families in which their PIs show different affinity to different proteases or different protease types. In the last 20 years, a significant number of new PI families have been identified, enlarging the number of families initially described by Laskowski and Kato [19]. However, some of them have not been characterized in detail yet. Also, the methods used for sequence and conformational structure analysis are continually under revision [24]. Currently, the PIs have been grouped into 85 different families and these families have been grouped into 38 clans when considering the classification system proposed by Rawlings et al. [3], the serine protease inhibitors are the most widely studied [1,25].

## 3. Mechanisms of Inhibition of Protease Inhibitors

The mechanisms of protease-inhibitor interaction were intensely revised by several authors [22,26,27]. Inhibitors can interact with proteases in different ways, although there are two mechanisms of interaction widely distributed in nature [3]. One of them is the irreversible trapping reaction and the best-characterized families of protease inhibitors that showed this mechanism correspond to the families of serpins (I4), α2 macroglobulins (I39) and baculovirus protein p35 inhibitors (I50) [3,23]. In this type of inhibition mechanism, the protease–inhibitor interaction induces the cleavage of an internal peptide bond in the inhibitor structure, triggering a conformational change (Figure 1A). This reaction is not reversible, and the inhibitor never recovers its initial structure. For this reason, the inhibitors that participate in trapping reactions are also known as suicide inhibitors. The other mechanism generally observed of protease–inhibitor interaction is known as a tight-binding reaction. This mechanism is also called a standard mechanism and it was extensity described by Laskowski and Qasim [28], and most recently by Farady and Craik et al. [29]. All inhibitors that operate by this mechanism are canonical and it was demonstrated for serine protease inhibitors [3]. The majority of plant serine protease inhibitors (SPIs) adopt the standard mechanism of inhibition [26]. In tight-binding reactions, the inhibitors interact with the protease active site (P1) in a similar way to the enzyme-substrate interaction (Figure 1B). The protease-inhibitor complex co-exists in a stable equilibrium among the intact form of the inhibitor and the modified forms of the inhibitor where the peptide bond of the reactive site is cleaved. Therefore, the inhibitor in the complex is dissociated to its intact or its modified form. The canonical inhibitors can also inhibit serine proteinases with differing P1 specificities. The Bowman Birk, Potato II and Kunitz families are able to target more than one proteinase at a time, often with different specificities [27]. Plant SPIs that apply this strategy allow plants to prepare themselves against unwanted proteolytic activity, whether to control development or to defend against pest attack. 

## 4. Plant Serine Protease Inhibitors (SPIs)

Plant SPIs include an abundant variety of proteins and play a fundamental role in several physiological processes. Plant SPIs represent between 1% and 10% of the total proteins in storage organs such as seeds and tubers, being able to inhibit different kind of serine proteases [18,21]. Most of the SPIs studied in the plant kingdom derivate from three main families named Leguminosae, Solanaceae and Gramineae [30]. Many studies have reported that SPIs found in other families showed a widespread distribution of SPIs throughout the plant kingdom [14,16,31]. These SPIs are classified as Bowman–Birk serine protease inhibitors, cereal trypsin/α-amylase inhibitors, mustard trypsin inhibitors, potato type I protease inhibitors, potato type II protease inhibitors, serpins, Kunitz type inhibitors, and squash serine inhibitors [26,28,32]. While the majority of SPIs specifically inhibit a type of proteases, some of them may act as multifunctional inhibitors [33]. Likewise, numerous studies provide a general description of the characteristics of the SPI families already known in plants [11,34], or describe the role of SPIs in a specific area of interest, such as pathogen responses [15,35,36]. In addition, evolutionary and global analysis of different SPI families from plant species and algae have recently been published [1,37]. Nevertheless, the role of most of the plant SPIs remain unknown.

In general terms, plant SPIs regulate the activity of proteases in two different situations: regulating the activity of endogenous proteases to control and avoid an indiscriminate degradation when it is not convenient [18,38]. For instance, the development of senescence corresponds to increased serine protease activity due to a reduction in SPI activity against serine proteases. Therefore, the delay of the senescence depends on a precise mechanism assured by a connection between the protease and protease inhibitor activities [39,40]. Alternatively, SPIs act to regulate the activity of exogenous proteases, as the proteases from certain plant pathogens such as insects, fungi, bacteria and parasites to prevent cellular damage [10,16,17]. As an example, SPIs from different families over-expressed in several plant species have conferred resistance to lepidopteran, coleopteran, homopteran [41] and acari [16,42]. Serpins and Kunitz type PIs are two interesting examples that have been extensively studied [43,44]. Several studies showed that *Arabidopsis* serpins (AtSerpin1, AtSerpin2, and AtSerpin3) are responsible for cell death and pathogen-associated stress [43,45]. In addition, it was demonstrated that AtSerpin1 acts in leaves during senescence [46]. In the same way, recent studies showed that a Kunitz type protease inhibitor called AtWSCP has an important role in the regulation of protease activity during plant development, especially in skotomorphogenesis and flower development [47,48]. Likewise, several SPIs from other families have been involved in the regulation of plant endogenous serine protease activity in various organs during seed germination, development and storage of proteins during dormancy [11,46,49]. On the other hand, under adverse conditions such as water deficit, the hydrolytic protein degradation in leaves is controlled by the accumulation of specific inhibitors like SPIs assuming that serine protease activities are regulated by their respective inhibitors [50,51,52].

In summary, plant SPIs are involved in the mobilization of storage proteins, regulation, and stabilization of endogenous enzymatic activities, morphogenesis, flower development, modulation of apoptosis, and cell death and in plant defense mechanisms against animals, insects, and microorganisms. Although studies on plant SPIs are mainly considered of great interest from basic science, recent evidence suggests that SPIs could have important applications. It is due to novel biotechnological developments using transgenic plants over-expressing SPIs that showed greater resistance to pests and diseases.

## 5. SPI as Protein Defense in Plants

The immune system of plants can be triggered in different situations. DAMPs (damage-associated molecular pattern)-triggered immunity (DTI) and MAMPs (microbe-associated molecular patterns)-triggered immunity (MTI) protects plants from a wide range of microbes [53] since this mechanism could be activated by the recognition of D/MAMP present in the pathogens [54,55]. During the evolution, certain pathogens have acquired the ability to suppress D/MTI by producing effector molecules, released by the pathogen. In response to this, plants have developed defense mechanisms called effector-triggered immunity (ETI) against these effectors [54,56,57]. Both D/MTI and ETI activate a cascade of signaling events involving receptors, MAP-kinases, hormones, and transcription factors, which develop resistance against the microorganism in coordination [54]. Different plant proteins, including protease inhibitors, lectins, ribosome inactivating proteins, and certain enzymes are involved in the protective barrier in the early stages of different types of pathogen infection [55]. Several reports showed that SPIs participate in plant defense against pests and phytopathogenic microorganisms. In addition, several phytopathogens secrete different types of extracellular proteases [58], which play an active role in the disease development [10,16,17]. Indeed, recent results showed that pathogen-secreted proteases activate the plant immune pathway through the MAPKs signaling cascade, involving the Gα, Gβ and Gγ subunits of heterotrimeric G-protein complex [59]. It is now known that plant SPIs are capable of inhibiting the extracellular serine proteases produced by phytopathogenic microorganisms, which are necessary to invade plant cells and to supply nutrients [8]. In response to the action of the extracellular proteases released by pathogens, plants induce the expression of SPIs to suppress the growth of microorganisms and control the infection process [10,60]. It is also known that plant SPIs are capable of suppressing the enzymatic activity in the digestive tract of insects, preventing the assimilation of vegetable proteins [8,61]. 

The relationship between serine proteases and SPIs secreted by pathogens and/or host plants has been extensively studied in the ETI [62,63]. However, little is known about the role of plant SPIs during the M/DTI and in the innate immune response [63]. It was observed that herbivory-induced JA accumulation activates the biosynthesis of anti-herbivore compounds, such as SPIs [64,65]. Remarkably, plant SPIs are one of the main anti-nutritional components induced by wounding because they interfere with the digestive system of herbivores, limiting their growth and development. In these cases, the activation of SPI genes is induced by jasmonic acid and this activation occurs not only in the wounded leaves but also in the distal leaves [8,66]. In fact, the up-regulation of SPI-encoding genes in response to biotic stress, mechanical wounding, herbivory, and jasmonates was demonstrated in several plant species, reinforcing their defensive roles [67,68]. Interestingly, Qu et al. [69] identified a novel Bowman–Birk type inhibitor (BBI) family in rice. Some members of this family could have a defensive function since these BBI genes expression is up-regulated in response to wounding. Indeed, the over-expression of one of them along with jasmonic acid increases the resistance to *Magnaporthe grisea*, a fungal pathogen causing rice blast. 

Consistently with the role of SPIs in the defense plant response, significant accumulation of SPIs occurs in certain tissues and organs which are likely to be attacked. In this sense, tissues or storage organs with high nutritional value for insect or pathogenic microorganisms, like seeds and tubers, are the main targets [70,71]. In fact, these tissues have been extensively studied describing multiple SPIs that belong to different families and clans [72]. The second attractive site for infecting are the fluids like phloem, where the pathogen is able to be transported along the whole host. Yoo et al. [73] showed that the concentration of serpins in the phloem of *Cucurbita maxima* is increased upon challenge with the aphid *Myzus persicae*, consistently with the role of SPIs in defense signaling. La Cour Petersen et al. [74] demonstrated that phloem serpins are mobile (graft transmissible) through grafting experiments using pumpkin and cucumber (*Cucumis sativus*) plants, suggesting that they are potential signaling molecules involved in the regulation of programmed cell death and/or defense pathways. An alternative function for serpins in the phloem may be to maintain the integrity of important signaling peptides or proteins through inhibition of destructive proteinase activity [35]. In insects, the PIs present in the hemolymph undoubtedly participate in the immune response, by regulating the prophenoloxidase activation in response to the pathogen invasion [75]. However, it has not been clarified whether several SPIs present in the phloem of aerial organs have a role against pathogens. 

## 6. Plant SPIs: Biotechnology Application in Agriculture

Plant SPIs are of particular interest because they act as protective agents codified by a single gene and inhibit proteolytic enzymes from animal and fungi, but rarely from plants [76,77]. The biotechnological potential of SPIs employed as protective agents has been demonstrated by transferring SPI genes from different sources to several plants of economic interest and the resulting transgenic plants are more resistant to pests [78,79] and pathogens [69,80,81] (Table 1). The first clue that SPIs would have a possible role in plant protection was observed when certain insect larvae were unable to develop normally in soybean products. Then, it was demonstrated that the trypsin inhibitors present in the soybean were toxic and they were responsible for the effect caused on the growth of larvae from flour beetles (*Tribolium confusum*) [9]. These findings suggest that plant SPIs can go into the insect digestive tract along with the food and block protein digestion, and the insect is then unable to absorb the nutrients leading to the retardation of its growth and development [8]. As it was described for some insects, plant SPIs (soybean Kunitz and Bowman–Birk inhibitors) would get access to the proteases present in the insect gut [82,83]. After these first studies, numerous plant SPIs have been characterized for their potential to control herbivorous insects. These studies included both *in vitro* assays to observe the effect of SPIs on the proteases present in the midgut of insects and *in vivo* bioassays by using artificial diets containing the purified inhibitors [8,9,68,84,85]. More recent *in vitro* assays showed that recombinant Kunitz trypsin inhibitor from poplar (*Populus* spp.) and soybean (*Glycine max*) differentially inhibited midgut proteases from *Mamestra configurata* and *Malacosoma disstria*, lepidopteran pests from poplars and crucifers, respectively [86]. In addition, Botelho-Junior et al. [87] showed that seven inhibitors (20–25 kDa) of the Kunitz type family purified by chromatography from the tropical crop passion fruit (*Passiflora edulis* Sims) and used in artificial diets displayed activity against midgut serine and cysteine proteases from the sugarcane borer *Diatraea saccharalis* and the coleopteran *Callosobruchus maculatus*. Another member of the Kunitz family, ApKTI, from seeds of the leguminous tree *Adenanthera pavonina* was able to inhibit simultaneously both trypsin and papain proteases [88] and it was active against gut proteases from herbivorous insects, including beetles and moths [89]. These authors demonstrated that artificial diets containing ApKTI reduced the viability and fertility of these insects, indicating that ApKTI acts in defense against several herbivorous insects. Recently, numerous studies have demonstrated the efficacy of protease inhibitor over-expression in transgenic plants which cause an increase in resistance to insect pest [76,90,91,92]. The first PI gene from the Bowman–Birk family successfully over-expressed in transgenic tobacco was the cowpea trypsin inhibitor gene (*CPTI*) [93]. Later, the *CPTI* gene was inserted in the genome of other plants like cotton, rice, cabbage, strawberry, sweet potato, potato or pigeon pea enhancing the resistance to different lepidopteran species [16]. Interestingly, more recently, Kunitz trypsin inhibitors from *A. thaliana* (AtKTI4 and AtKTI5) transiently expressed in *Nicotiana* plants showed their bifunctional features to inhibit cysteine- and serine-proteases present in the midgut interfering in the correct hydrolysis of dietary proteins [48]. On the other hand, different SPIs were obtained from crop plants, such as rice, barley, soybean cowpea, sweet potato and maize and were overexpressed in several plant species conferring resistance to several species of insect pests [94,95,96,97,98,99,100]. In particular, the co-expression of potato type I and II proteinase inhibitors from *Solanum tuberosum* (StPin1a) and *Nicotiana alata* (NaPI), respectively, provided protection against insect damage in cotton crops [101]. Besides, different serine and cysteine proteinase inhibitors have been identified in barley. In particular, the barley trypsin inhibitor CMe (BTI-CMe) which belongs to the family of protease/α-amylase inhibitors, showed a high inhibition of trypsin-like activity and it has been successfully used to improve resistance toward different pests [94,101,102]. Transgenic expression of BTI-CMe in indica and japonica rice, conferred resistance to the rice weevil *Sitophilus oryzae* [101]. Besides, the insect resistance in transgenic wheat stably expressing BTI-CMe was increased [94]. The expression of two barley PIs (BTI-CMe and a cysteine proteinase inhibitor, Hv-CPI2) in tomato promoted endogenous defense response and enhanced resistance against *Tuta absoluta* [102]. These two PIs showed an additive effect and better efficiency was achieved when both genes were co-expressed. Importantly, transgenes expression had no harmful effect on *Nesidiocoris tenuis* (Reuter) (Heteroptera: Miridae), a predator of *T. absoluta* [102]. These results show the versatility of plant SPIs to inhibit protease belonging to different families and the high potential to control the herbivorous insects from different species.

On the other hand, numerous studies were performed to demonstrate the inhibitory capacity of SPIs on the fungi and bacteria growth. Dunaevsky et al. [104] showed that a trypsin/chymotrypsin inhibitor (BWI-1a) from buckwheat (*Fagopyrum sculentum*) was able to interfere with the spore germination and mycelial growth of the tobacco fungus, *Alternaria alternata*. It was also demonstrated that several chymotrypsin inhibitors belong to Kunitz type family and isolated from potato tubers were able to interfere with the growth and development of the oomycetes of *Phytophthora infestans* [105]. Also, Ye et al. [106] demonstrated that a Bowman–Birk-type trypsin-chymotrypsin inhibitor from *Vicia faba* suppressed the growth of mycelia from different types of fungi suggesting its broad-spectrum capacity of inhibition. Similarly, Pekkarinen et al. [107] showed that three SPIs (chymotrypsin/subtilisin inhibitor 2, alpha-amylase/subtilisin inhibitor and Bowman–Birk trypsin inhibitor) from barley are capable of inhibiting various serine proteases from the fungus *Fusarium culmorum*. More recently, Laluk and Mengiste [91] identified an unusual SPI (UPI) from *A. thaliana* that was involved in the defense against the fungi *Botrytis cinerea* and *Alternaria brassicicola*. Similar results were observed by Pariani et al. [108] where the authors characterized two putative Kazal-type inhibitors from *A. thaliana*. They showed that these inhibitors exhibit a strong antifungal activity by inhibiting *in vitro* the germination rate of *B. cinerea* conidia. All these reports show that SPIs are able to inhibit not only proteolytic enzymes from parasites, fungi and/or bacteria, but also serine proteases from the digestive tract of insects, suggesting that the transference of a single proteinase inhibitor gene from different sources to plants of economic interest would result in obtaining transgenic plants more resistant to a great diversity of pathogens [80,92,93,115,116,117]. On the other hand, the co-expression of different types of SPIs could benefit the plant resistance due to the synergistic effect among them. Finally, the inhibition capacity present in the SPIs showed that they are an attractive strategy to be implemented in the control of pests and other pathogens (Table 1).

## 7. Challenges and Perspectives in Pathogen Resistance

The great advances in modern agriculture have allowed the progressive elimination of exogenous pesticides towards the use of practices more sustainable and environmentally benevolent. During this decade, a significant proportion of transgenic crops has been engineered to express pathogen resistance. However, no commercial transgenic product with enhanced disease resistance is currently available, except for *Bacillus thuringiensis* toxin genes [103]. Since the capacity for pest controlling is often partial and resistance is only effective against specific pest biotypes [118], alternative strategies should be developed to use novel genes encoding anti-microbial and insecticidal products with suitable characteristics for applying in transgenic crops. In this context, the use of SPIs can be an attractive system demonstrated by the vast bibliography reviewed in this paper and in others [18]. However, there are some aspects that should be considered for this technology to be successfully incorporated. One of them is the adaptation ability of pathogens to express proteases that may not be recognized by plant PIs yet. Another point to be considered is the co-evolution between certain PI gene families and their pathogen counterpart [69]. In fact, a large number of insect pests and pathogenic microorganisms have evolved and are adapted to the host plant PIs, thus transgenic crops did not exhibit enhanced resistance to them [119,120]. By contrast, other studies have shown that PIs from nonhost plants can efficiently inhibit the proteases from crop pests [121]. Therefore, it is important to assure that the resistant transgenic plant does not promote the development of resistance in pathogens. In this sense, novel resistant transgenic plants should be drafted in with the aim of delaying/preventing the onset of the infection and provide more durable levels of crop protection [122]. To achieve this goal, it is important to identify the SPIs that have evolved separately from the crop pathogens. With the advent of functional genomics, the discovery of new SPI genes and their function in plant processes opens the opportunity to generate transgenic plants based mainly on multigenic traits. In this sense, it is important to understand more completely the role of SPIs in the innate and acquired plant immune responses.

Up until now, the translate pathway in response to pathogens and its relationship with the role of SPIs are not completely elucidated and this knowledge will be crucial for controlling pathogens when using transgenic crops over-expressing SPIs. As far as we know, there is no information about plant receptor-pathogen protease interactions or direct linkage between them during the induction of plant defense pathways. Future experiments would allow us to fully understand the repertoire of pathogenic proteases, the pattern of transcriptional changes in gene expression of plant-induced SPIs, and the SPI expression response during the plant growth or infection. These experiments could become the basis to establish novel strategies involving the use of SPIs as a tool for pest and pathogen control. Undoubtedly, all the knowledge related to the role of SPIs during plant-pathogen interaction will contribute to the development of more efficient and environmentally friendly pest control strategies. In this sense, the current sequencing of crop genomes, together with comprehensive gene expression and functional gene analysis, will boost the development of transgenic disease-resistant plants. In particular, strategies using general defense pathways and antifungal protein overexpression like SPIs will benefit the crops, since they will have a broad spectrum of defense arsenal. However, it should be mentioned that the public concern about the use of transgenic crops and its impact on the environment increases daily [123]. Although the defenders of transgenic plants claim that crops are environment-friendly, do not pose a risk to human health, and are profitable for farmers; the detractors are still arguing that transgenic crops can be injurious to human and animal health because they have not been properly tested. In this context, new biotechnology alternatives should be developed. Two new high-throughput genome editing technologies, Transcription Activator-like Effector Nucleases (TALENs) and CRISPR/Cas system, are being used as tools for the study of plant pathogenesis and adaptive immunity. Both technologies allow the modulation of gene expression in plants and/or the development of disease-resistant plants [124]. These technologies can be employed for modulating the expression of endogenous plant SPIs that have demonstrated efficacy on pathogen resistance. The application of these technologies as new tools for pest, fungi and bacteria control would reduce the use of transgenic crops and will allow new resistant sources of important crops to be obtained to ensure a healthy environment and public approbation (Figure 2).

## 8. Plant SPIs: Biotechnology Application in Molecular Farming

Molecular farming refers to the production of recombinant proteins in plants, including pharmaceutical products, industrial proteins, and other secondary metabolites. Between 1986 and 1989, the first pharmaceutic product (the human growth hormone) and the first recombinant antibody were expressed in transgenic plants [125]. However, it was not until 1997 that the chicken egg avidin protein was commercialized as the first recombinant protein produced in plants [126]. These findings demonstrated that plants can be employed as platforms to produce large-scale recombinant proteins. Over the years, it was demonstrated that plants have the capability to express functionally active proteins from mammals and other eukaryotic organisms with therapeutic activity like human sera, growth factors, vaccines, hormones, cytokines, enzymes and antibodies [127]. This is possible due to the ability of plants to perform the post-translational modifications required for the correct folding of the exogenous proteins in order to keep their functionality and integrity [128,129]. Thus, there is great interest in using plants as bio-factories to produce drugs, antigens, nutritional supplements, biopolymers and biofuels [130]. Despite plant production systems possess great advantages over other systems already established for recombinant proteins production (yeast and animal cells), there are some limitations that make plants less socially accepted [131,132]. One of the most important challenges for the science community that use plants as a commercial productive platform is to improve the yields of recombinant proteins expressed within them [130,133]. In this sense, to explore new strategies for minimizing the degradation of foreign proteins may contribute to increasing the commercial potential of pharmaceutical, industrial interest or vaccines produced in plants [134,135].

Degradation of recombinant proteins by proteases due to imperfect synthesis or deficient folding can strongly affect the protein accumulation levels [136]. In addition, proteases may affect the integrity of recombinant proteins not only altering the protein production but also their biological activity during the extraction steps [109,137,138,139]. The proteomic data for *A. thaliana*, rice, *N. tabacum,* and *N. benthamiana* allowed us to identify the different protease families present in several subcellular compartments [135]. Proteases are more abundant in the vacuole, chloroplast, and apoplast [62,140,141,142]. The knowledge of protease profile in cell compartments where the recombinant protein would be expressed has contributed to deciding whether an inhibitor or a combination of inhibitors might work the best.

Several studies have demonstrated that the co-expression of PIs in transgenic plants or in plants that transiently express recombinant proteins have contributed to minimizing the proteolysis of recombinant proteins, is an economically viable option [135]. Given the wide versatility of plant IPs, they comprise an interesting alternative to diminish the degradation of recombinant proteins expressed in plants by protease activity (Figure 3). In particular, PIs with specificity against proteases such as cysteine, serine or aspartic proteases have been employed for this purpose [110,111,143] (Table 1). However, since serine protease activity is the most active protease in plant cells, most of the works have focused on SPIs activity against chymotrypsin and trypsin-like proteases [135]. In addition, given that the apoplast is the default destination for antibodies, many studies have evaluated how to control proteolysis throughout the secretory pathway. Then plant protease inhibitors have been co-expressed during their migration through the cell secretory pathway to protect secreted proteins. In this sense, Komarnytsky et al. [113] demonstrated that co-secretion of a plant Bowman–Birk type protease inhibitor was able to reduce the degradation of immunoglobulin complexes in the secretion pathway increasing the antibody production. This study initially demonstrated that the use of protease inhibitor as a companion protein achieves high yields of complicated therapeutic proteins secreted from plant roots. Similarly, Kim et al. [114] engineered a synthetic chymotrypsin and trypsin inhibitor from *N. alata* to reduce the extracellular protease activity in the suspension culture medium. They showed that the co-expression of this SPI enhanced the accumulation of the recombinant human granulocyte–macrophage (hGM-CSF) in transgenic rice cell suspension culture. Additionally, this study showed that the expression of this plant protease inhibitor did not affect the plant growth and development. In addition, Rivard et al. [138] demonstrated that the over-expression of two SPIs (tomato cathepsin D inhibitor (CDI) or bovine aprotinin) in potato plants improved the stability of proteins in leaf crude extracts during the extraction/recovery process. More recently, Goulet et al. [110] showed that aspartic and serine protease inhibitors are the main modulators of protease activities in the apoplast of *N. benthamiana* leaves. Therefore, they transiently expressed two broad-spectrum inhibitors (apoplast targeted versions of tomato cathepsin D and tomato cystatin inhibitors) in tobacco leaves. The results showed that the transient co-expression of both protease inhibitors increased the recombinant murine antibody accumulation by 70%–80% along the leaf cell secretory pathway [110]. Similar results were obtained by Robert et al. [111] and Grosse-Holz et al. [112]. In both studies, the co-expression of PIs by agroinfiltration improved the yield of pharmaceutical recombinant proteins in *N. benthamiana* leaves. In fact, these approaches can result in up to 40% yield improvement per plant persisting in the oldest leaves where proteolytic activities are stronger [111]. Taken together, all these results suggest that these plant broad-spectrum PIs are effective companion proteins for the *in planta* protection of recombinant proteins transiently expressed in leaves (Table 1). On the other hand, compared to prokaryotic systems, natural protease-deficient plants do not exist for plant species currently useful for recombinant protein production. To address this problem, Pillay et al. [136] evaluated the use of transgenic tobacco plants expressing the rice cysteine protease inhibitor oryzacystatin-I to increase the accumulation and activity of the *Escherichia coli*-derived glutathione reductase. The authors showed that the lower cysteine protease activity was directly related to higher glutathione reductase activity and higher glutathione reductase amounts. In addition, recent evidence showed that the expression in leaves of protease inhibitors may have a positive impact on protein levels, with insignificant effects on growth and plant development [109,144]. These results suggest that the draft of a transgenic plant expressing PIs would provide a suitable cellular environment depleted of protease activity and could improve the transient expression platform using tobacco plants.

## 9. Challenges and Perspectives in Preventing Proteolysis in Plant Protein Factories

There is a necessity to develop strategies to prevent proteolytic degradation of plant-expressed recombinant proteins, so it is important to understand when and where this degradation occurs. Several studies have demonstrated that the levels of recombinant protein production in plants is significantly improved when they are accumulated in the endoplasmic reticulum (ER) since this compartment provides a protective oxidizing environment along with molecular chaperones useful for their correct protein folding, low protease activity and room enough for protein accumulation [145,146,147,148]. However, ER lumen is not always considered a suitable destination because several therapeutic proteins, especially the antibodies, require post-translational modifications for their stability and biological activity and this occurs downstream of the ER, along the secretory pathway [149]. Thus, protein targeting to apoplast may help to correct maturation and glycosylation of recombinant proteins [150]. In addition, several findings suggest that degradation of recombinant proteins expressed in a plant is not related to the extraction process, but along the secretion into the apoplastic space [151]. Therefore, it is important to gain more knowledge of different proteases mainly present in the apoplast in order to develop strategies that would contribute to preventing unintended proteolytic processing and help to the correct the maturation of recombinant proteins [152]. Although recent studies identified different protease families present in the apoplast space of tomato, *A. thaliana*, *N. benthamiana* and *N. tabacum* [62,153], they require more detailed analyses to confirm their direct involvement in recombinant protein degradation. This information will be useful for planning strategies before recombinant protein production [151]. It will also allow establishing smart protocols combining the introduction of PIs of specific serine, cysteine, aspartic acid, and/or metalloprotease according to the requirements to increase the heterologous protein production. On the other hand, one of the strategies with the greatest potential in molecular farming platforms is the tobacco leaf agroinfiltration. This technology is being widely used by several research and development (R&D) groups to produce interesting proteins in the pharmacology field [130,131]. In this sense, great advances have been made in the identification of the proteases involved in recombinant protein degradation, particularly in *Nicotiana* species. Most of the protease families, which cooperate in recombinant protein production in *Nicotiana* species, belong to the aspartic and cysteine protease (papain-like) families and, to a lesser extent, the serine and metallo-protease families [110,153]. In particular, Goulet et al. [110] described that the *N. benthamiana* leaves contain less protease activity than *N. tabacum* leaves, suggesting that the first would be more suitable to be used for agroinfiltration. Also, the authors observed that the proteolysis level is higher after agroinfiltration. Therefore, the identification of proteases induced by agroinfiltration could be a key step in the improvement of recombinant protein production when applying the agroinfiltration technique. Consequently, searching for plant species with low proteolytic activity will provide improved recombinant protein stability and the screening of plant species with low protease activity is, therefore, urgently required. In fact, Santos et al. [154] showed that the number of proteases found in Medicago was considerably lower. Genomic and proteomic approaches will allow us to advance in the knowledge of the specific role of the different proteases and their inhibitors and this will result in the development of new strategies to improve plant-based recombinant protein production significantly. Although each protein has specific characteristics, and this makes necessary to test different strategies, it is expected that the knowledge of proteolytic processes and protein maturation in plants will allow us to have a set of biotechnology tools to improve the pharmaceutical protein production.

## 10. Conclusions

Several transgenic plants that express SPIs have been produced and tested in order to increase the resistance against pathogenic organisms. In addition, SPIs and other PIs from different families have been used to minimize the proteolysis of recombinant proteins expressed in plants. Several successful examples have been mentioned in this review and many of them have comprehensive perspectives to be implemented in the molecular agriculture fields. Coordinated efforts in both areas to develop eco-friendly strategies for protecting plants against pests and pathogens with the added value of improving the plant’s overall fitness will enhance the commercial value of the plant platform for protein production.

## Figures and Tables

**Figure 1 ijms-20-01345-f001:**
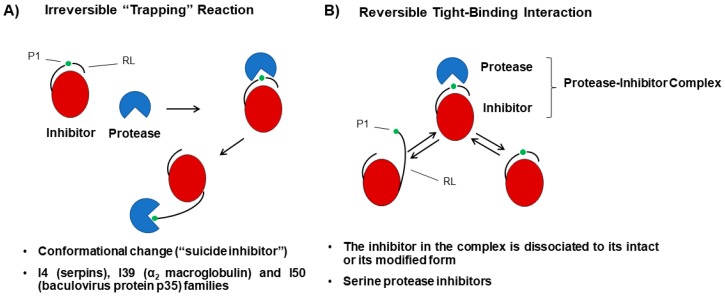
Mechanisms of protease-inhibitor interactions. (**A**) Irreversible “trapping” reactions. The protease–inhibitor interaction induces the cleavage of an internal peptide bond in the inhibitor triggering a conformational change. This reaction is not reversible, and the inhibitor never recovers its initial structure. For this reason, the inhibitors that participate in trapping reactions are also known as suicide inhibitors. The inhibitors never recover the initial structure. (**B**) Reversible tight-binding interactions. The inhibitor interacts with the protease active site in a similar way to the enzyme-substrate interaction. The protease-inhibitor complex co-exists in a stable equilibrium among the intact form of the inhibitor and the modified forms of the inhibitor where the peptide bond of the reactive site is cleaved. Therefore, the inhibitor in the complex is dissociated to its intact or its modified form. P1: PI reactive site; RL: reactive loop.

**Figure 2 ijms-20-01345-f002:**
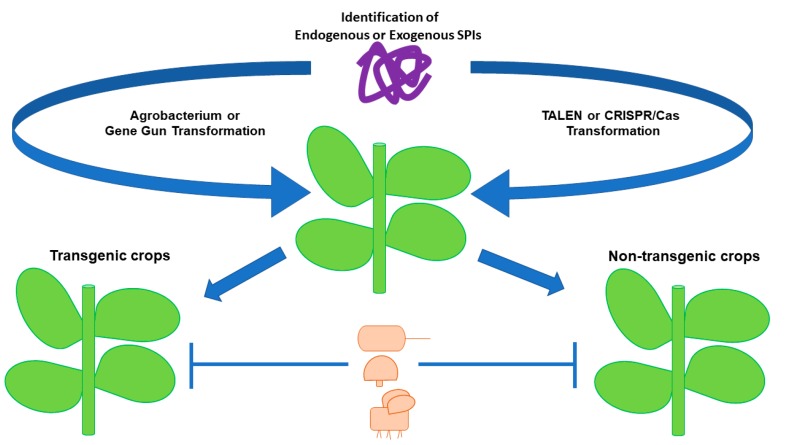
Serine proteases inhibitors identified in plants (endogenous SPIs) or in other organisms (exogenous SPIs) can be introduced by conventional transformation (*Agrobacterium tumefaciens* or gene gun transformation) or by novel editing technologies to increase the resistance to insect pest and phytopathogenic microorganisms. The application of these technologies can be used to produce new resistant sources of important crops.

**Figure 3 ijms-20-01345-f003:**
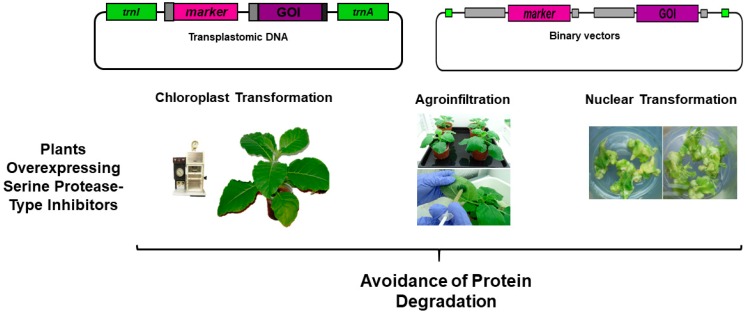
Co-expression of serine protease inhibitors could help to minimize the proteolytic activity and to avoid the recombinant protein degradation in transplastomic, transgenic plants or in plants that transiently express recombinant proteins. The protease inhibitor can be targeted in the same organelle (chloroplast transformation) where the recombinant protein would be expressed or co-expressed. Transgenic plants over-expressing protease inhibitors would be more suitable to express recombinant proteins by agroinfiltration.

**Table 1 ijms-20-01345-t001:** Plant protease inhibitors with potential application in agriculture and molecular farming.

SPI Name	Origen	Role and Function	Biotechnology Application	References
*A. thaliana* Kunitz trypsin inhibitors (AtKTI4, AtKTI5)	*Arabidopsis thaliana*	Inhibitory activity against serine and cysteine protease; effect on mite performance (fecundity and mortality)	Protection against spider mite	[42]
AtSerpin1	*Arabidopsis thaliana*	Inhibition of digestive protease activity; inhibition of larval growth; inhibition of RD21 activity	Protection against insect disease	[47,103]
Kunitz type protease inhibitor (AtWSCP)	*Arabidopsis thaliana*	Inhibition of cysteine RD21 activity; controlling cell death	Protection against herbivore attack	[45,47]
Potato type 1 inhibitors	*Solanum tuberosum*	Differential expression pattern after wounding and nematode infection	Protection against nematodes	[68]
Bowman-Birk-type inhibitor	*Oryza sativa*	Arrest fungal invasion; inhibition of fungal growth	Protection against fungal disease	[69]
Phloem serpin-1 (CmPS-1)	*Cucurbita maxima*	Inhibition of elastase activity; increase of the aphid mortality	Protection against insect disease	[73,74]
Cowpea trypsin inhibitor gene (CPTI)	*Vigna unguiculata*	Inhibition of larval growth	Protection against insect disease	[78,79,93]
Potato carboxypeptidase inhibitor (PCI)	*Solanum tuberosum*	Antifungal activity; inhibition of larval growth	Protection against fungal and insect disease	[80,81]
Maize proteinase inhibitor (mPI)	*Zea mays*	Inhibition of digestive serine proteinases; inhibition of larval and fungal growth	Protection against fungal and insect disease	[81,96]
Soybean Kunitz inhibitor (SKTI)	*Glycine max*	Inhibition of digestive proteases present in insects and parasites	Protection against parasitic and insect disease	[83,86,95]
Soybean Bowman-Birk inhibitor (SbBBI)	*Glycine max*	Inhibition of digestive protease activity; inhibition of aphid growth	Protection against aphid parasitoids	[83]
Poplar Kunitz trypsin inhibitor	*Populus trichocarpa* x *Populus deltoides*	Inhibition of midgut protease present in lepidopteran pests	Protection against insect disease	[86]
Passion fruit Kunitz type inhibitors (PfKI)	*Passiflora edulis* Sims	Inhibition of midgut proteases present in lepidopteran and coleopteran pests and *Aedes aegypti*	Protection against insect disease and Control of vectors of neglected tropical diseases	[87]
Kunitz trypsin inhibitor (ApKTI)	*Adenanthera pavonina*	Inhibitory activity against trypsin and papain proteases; inhibition of midgut proteases and larval growth	Protection against insect disease	[88,89]
Unusual serine protease inhibitor (UPI)	*Arabidopsis thaliana*	Chymotrypsin inhibitory activity; effect on the fungal and larval growth	Protection against fungal and insect disease	[91]
Serine proteinase inhibitor (BvSTI)	*Beta vulgaris*	Trypsin inhibitor activity; effect on larval weights	Protection against lepidopteran insect disease	[92]
Serine protease inhibitor CMe (BTI-CMe)	Barley (*Hordeum vulgare*)	Inhibition of midgut protease activity; effect on larval growth and survival of insects	Protection against insect disease	[94,101,102]
Potato type I (StPin1A) inhibitor/Potato type II (NaPI) inhibitor	*Solanum tuberosum* *Nicotiana alata*	Protease inhibitory activity; effect on larval growth	Protection against *Helicoverpa* spp.	[97]
PI-I and PI-II-class inhibitors	*Solanum nigrum*	Serine protease inhibitory activity	Protection against insect disease	[98]
Potato Type II Proteinase Inhibitors (SaPIN2b)	*Solanum americanum*	Inhibition of midgut protease activity	Protection against insect disease	[97,100]
Serine protease inhibitor (BWI-1a)	*Fagopyrum sculentum*	Inhibition of spore germination, mycelial growth, bacterial growth and survival of insects	Protection against insect, fungal and bacterial disease	[104,59]
Serine protease inhibitors (PSPI-21, PSPI-22)	*Solanum tuberosum*	Trypsin and chymotrypsin inhibitory activity; inhibition of mycelial growth	Protection against fungal disease	[105]
Bowman-Birk-type inhibitor	*Vicia faba*	Trypsin and chymotrypsin inhibitory activity; inhibition of mycelial growth	Protection against fungal disease	[106]
Chymotrypsin/subtilisin inhibitor 2, amylase/subtilisin inhibitor, Bowman-Birk trypsin inhibitor	*Hordeum vulgare*	Inhibition of subtilisin and trypsin proteases of *Fusarium culmorum*	Protection against fungal disease	[107]
Kazal type inhibitor (AtKPI-1)	*Arabidopsis thaliana*	Inhibition of conidial germination	Protection against fungal disease	[108]
Tomato cathepsin D inhibitor (CDI)	*Solanum tuberosum*	Improvement of the stability of proteins in leaf crude extracts	Achieves high yields of recombinant proteins in the extraction/recovery process	[109,110,111,112]
Bowman–Birk type protease inhibitor (BBI)	*Glycine max*	Reduction of the degradation of immunoglobulins in the secretion pathway	Achieves high yields of therapeutic proteins in transgenic plants	[113]
Chymotrypsin and trypsin inhibitor	*Nicotiana alata*	Reduction of the extracellular protease activity	Achieves high yields of recombinant proteins in cell suspension culture	[114]

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
