# Peer review of "Plant Serine Protease Inhibitors: Biotechnology Application in Agriculture and Molecular Farming"

_ijms, 2019, doi:10.3390/ijms20061345_

Reviewer 1 Report

In this review, the authors mainly summarized the importance of plant serine proteases inhibitors and their potential applications in the agricultural field. It is well-written, and will be helpful for the readers in the field of plant biotechnology. There are still, however, some suggestions and corrections with the current edition.  

Major Revisions:

1.    I suggest authors to tabulate all plant serine protease inhibitors discussed in this manuscript: like their origin, role and functions, application in agriculture etc..

2.    Figure 2 is not clear, modify it. Use same font as in the text in all the figures.

Minor revisions

1.    In all the figure legends are repeated. Delete Fig.1, Fig.2, Fig. 3 and Fig. 4 from figures. Check journal guidelines.

2.    Line 322: expand A. tumefaciens

3.    Line 279-329: split in to two paragraphs.

Author Response

Response to Reviewers:

Reviewer 1

Major Revisions:

1.    I suggest authors to tabulate all plant serine protease inhibitors discussed in this manuscript: like their origin, role and functions, application in agriculture etc.

Thank you very much for your suggestion. We made a table indicating the serine protease inhibitor’s name, origin, roles/functions, and application in biotechnology agriculture or molecular farming.

2.    Figure 2 is not clear, modify it. Use same font as in the text in all the figures.

According to reviewer’s #2 suggestion, the fig.2 was removed and the information was mentioned in the text of the manuscript.

Minor revisions

1.    In all the figure legends are repeated. Delete Fig.1, Fig.2, Fig. 3 and Fig. 4 from figures. Check journal guidelines.

It was checked.

2.    Line 322: expand A. tumefaciens

It was done.

3.        Line 279-329: split in to two paragraphs.

It was done.

Reviewer 2 Report

In this manuscript, the authors introduced and reviewed the classification and inhibition mechanisms of protease inhibitors (PIs), the functions of serine protease inhibitors (SPIs) in plant immunity and molecular farming. The review is timely and the data is interpreted ok.  The topic and content should be of interest to many readers of IJMS. However, before publication, the following questions and comments need to be addressed:

Line 9: The main function of SPIs as protease inhibitors

Introduction paragraph 1 and 2: This manuscript mainly discussed the plant SPIs. The initial studies of PIs may focus on animals, but the authors need to highlight these research results are in animals.

Line 32: which several biological processes?

Line 38-40: please cite the original research paper.

Line 65: a PI

Line 46-78 (section: Classification of protease inhibitors): The authors need to add the content about the identification and classification of plant protease inhibitors.

Line 91: most recently by Kelly et al. [25].?

Reference 25. Kelly, C.A.; Laskowski, M.; Qasim, M.A. The role of scaffolding in standard mechanism serine proteinase inhibitors. Protein Pept. Lett. 2005, 12, 465-471.

Line 79-107 (Section Mechanisms of inhibition of protease inhibitors including Fig. 1): I didn’t see any mechanism characterization and discussions of plant PIs examples, although the authors indeed cite some reference related to plant PIs (such as reference 22 and 23). The authors need to add the related content.

Moreover, are the authors sure that the mechanism characterization is most recent? Because the related references 24 and 25 are too old. Did the authors check the reference (Mechanisms Of Macromolecular Protease Inhibitors. doi: 10.1002/cbic.201000442) or other new ones?

Figure 2 are too simple. These contents can be described completely by one or two sentences. Can the authors add the related PI names and references corresponding to their roles in the figure 2? Or change the figure 2 to one table.

Line 113: Are the authors sure the demonstration is correct? Reference 27 was published in 1991. Most of the SPIs studied in the plant kingdom derivate from three main families named Leguminosae, Solanaceae and Gramineae [27]. ?

Line 114-117: These demonstrations are about PIs, not SPIs. Please double check the reference and revise these.

Line 132-138: Do the author have the reference or evidence for that: Kunitz-type protease inhibitor AtWSCP is a SPI?

Line 153: DAMPs (Damage-associated molecular patterns) -triggered immunity (DTI) and MAMPs (microbe-associated molecular patterns) -triggered immunity (MTI) protect plants …..

Also change the following PTI to MTI.

If the authors use PAMPs (pathogen-associated molecular patterns) -triggered immunity (PTI) to replace MAMPs (microbe-associated molecular patterns) -triggered immunity (MTI), then don’t need.

Line 185: Bowman–Birk type protease inhibitors (BBIs) are SPIs?

Line 206-277: Please check the references to clarify the classification of protease inhibitors in this section when the authors give the examples.

Author Response

Response to Reviewer 2

Line 9: The main function of SPIs as protease inhibitors

Introduction paragraph 1 and 2: This manuscript mainly discussed the plant SPIs. The initial studies of PIs may focus on animals, but the authors need to highlight these research results are in animals.

Line 32: which several biological processes?

It was clarified.

Line 38-40: please cite the original research paper.

These two cites were included: Mosolov, V.V.; Valueva T.A. Proteinase Inhibitors and Their Function in Plants: A Review. App. Biochem. Microbiol. 2005, 41, 227–246. Valueva, T.A.; Mosolov, V.V. Role of inhibitors of proteolytic enzymes in plant defense against phytopathogenic microorganisms. Biochemistry (Moscow). 2004, 69, 1305-1309.

If the reviewer does not agree with this, please indicate which are the original research papers that he/she would like to include.

Line 65: a PI

It was modified.

Line 46-78 (section: Classification of protease inhibitors): The authors need to add the content about the identification and classification of plant protease inhibitors.

It was added (lines: 55-60).

Line 91: most recently by Kelly et al. [25].?

Reference 25. Kelly, C.A.; Laskowski, M.; Qasim, M.A. The role of scaffolding in standard mechanism serine proteinase inhibitors. Protein Pept. Lett. 2005, 12, 465-471.

Kelly et al. 2005 was replaced by Farady and Craik, 2010 [in the revised version: cite 29].

Line 79-107 (Section Mechanisms of inhibition of protease inhibitors including Fig. 1): I didn’t see any mechanism characterization and discussions of plant PIs examples, although the authors indeed cite some reference related to plant PIs (such as reference 22 and 23). The authors need to add the related content.

Thank you very much for your suggestion, related contents to reference 22 and 23 were added (Lines: 98-108).

Moreover, are the authors sure that the mechanism characterization is most recent? Because the related references 24 and 25 are too old. Did the authors check the reference (Mechanisms Of Macromolecular Protease Inhibitors. doi: 10.1002/cbic.201000442) or other new ones?

Thank you very much for your suggestion, we have added Farady and Craik (2010) in the reference section instead of the reference 25. In addition, we have revised this article, and in general terms, the authors use competitive inhibition instead of standard inhibition and irreversible inhibition as the two mechanisms of protease inhibitors.

Figure 2 are too simple. These contents can be described completely by one or two sentences. Can the authors add the related PI names and references corresponding to their roles in the figure 2? Or change the figure 2 to one table.

Thank you very much for your suggestion, we have removed the Fig. 2. and the contents were described in the text of the manuscript.

Line 113: Are the authors sure the demonstration is correct? Reference 27 was published in 1991. Most of the SPIs studied in the plant kingdom derivate from three main families named Leguminosae, Solanaceae and Gramineae [27]?

It is true that reference 27 is too old. We have modified the sentence and new references were included. (lines: 126-128).

Line 114-117: These demonstrations are about PIs, not SPIs. Please double check the reference and revise these.

It was revised (lines: 130-131).

Line 132-138: Do the author have the reference or evidence for that: Kunitz-type protease inhibitor AtWSCP is a SPI?

Although it is true that AtWSCP is not able to inhibit serine protease, AtWSCP has a Kunitz-type protease inhibitor family motif. In that context, we have mentioned AtWSCP as an SPI (in the revised version we modified for AtWSCP PI). We consider AtWSCP as an interesting example to be mentioned in our review although it does not inhibit serine protease strictly. However, if the reviewer thinks that this information can be confusing, we can eliminate the mention of this inhibitor in the manuscript.

Line 153: DAMPs (Damage-associated molecular patterns) -triggered immunity (DTI) and MAMPs (microbe-associated molecular patterns) -triggered immunity (MTI) protect plants

Also change the following PTI to MTI.

If the authors use PAMPs (pathogen-associated molecular patterns) -triggered immunity (PTI) to replace MAMPs (microbe-associated molecular patterns) -triggered immunity (MTI), then don’t need.

It was modified (lines: 167-174).

Line 185: Bowman–Birk type protease inhibitors (BBIs) are SPIs?

We think that Bowman-Birk type protease inhibitors (BBIs) identified by Qu et al. (2003) belong to the SPIs. Despite the particular characteristics of this novel family of BBIs, we think that they can be included in the SPIs. However, if the reviewer considers that this novel BBI family is not SPI, we can eliminate any mention of it. Anyway, we modified the sentence indicating that it is a new family of BBIs identified in rice. (Lines: 200-201).

Line 206-277: Please check the references to clarify the classification of protease inhibitors in this section when the authors give the examples.

Thank you very much for your suggestion, we have checked the references and the information mentioned in the manuscript was clarified.

Round  2

Reviewer 1 Report

Authors well addressed all the comments suggested. In my opinion, the work can be accepted in current form for publication.

Reviewer 2 Report

The authors addressed my concerns.